# Skinning a Parameterization of Three-Dimensional Space for Neural Network Cloth

## Abstract

We present a novel learning framework for cloth deformation by embedding virtual cloth into a tetrahedral mesh that parametrizes the volumetric region of air surrounding the underlying body. In order to maintain this volumetric parameterization during character animation, the tetrahedral mesh is constrained to follow the body surface as it deforms. We embed the cloth mesh vertices into this parameterization of three-dimensional space in order to automatically capture much of the nonlinear deformation due to both joint rotations and collisions. We then train a convolutional neural network to recover ground truth deformation by learning cloth embedding offsets for each skeletal pose. Our experiments show significant improvement over learning cloth offsets from body surface parameterizations, both quantitatively and visually, with prior state of the art having a mean error five standard deviations higher than ours. Without retraining, our neural network generalizes to other body shapes and T-shirt sizes, giving the user some indication of how well clothing might fit. Our results demonstrate the efficacy of a general learning paradigm where high-frequency details can be embedded into low-frequency parameterizations.

## 1 Introduction

Cloth is particularly challenging for neural networks to model due to the complex physical processes that govern how cloth deforms. In physical simulation, cloth deformation is typically modeled via a partial differential equation that is discretized with finite element models ranging in complexity from variational energy formulations to basic masses and springs, see e.g. Baraff & Witkin (1998); Bridson et al. (2002; 2003); Grinspun et al. (2003); Baraff et al. (2003); Selle et al. (2008). Mimicking these complex physical processes and numerical algorithms with machine learning inference has shown promise, but still struggles to capture high-frequency folds/wrinkles. PCA-based methods De Aguiar et al. (2010); Hahn et al. (2014) remove important high variance details and struggle with nonlinearities emanating from joint rotations and collisions. More recently, Gundogdu et al. (2019); Santesteban et al. (2019); Patel et al. (2020); Jin et al. (2020) leverage body skinning Magnenat-Thalmann et al. (1988); Lander (1998); Lewis et al. (2000) to capture some degree of the nonlinearity; the cloth is then represented via learned offsets from a co-dimension one skinned body surface. Building on this prior work, we propose replacing the skinned co-dimension one body surface parameterization with a skinned (fully) three-dimensional parameterization of the volume surrounding the body.

We parameterize the three-dimensional space corresponding to the volumetric region of air surrounding the body with a tetrahedral mesh. In order to do this, we leverage the work of Lee et al. (2018; 2019), which proposed a number of techniques for creating and deforming such a tetrahedral mesh using a variety of skinning and simulation techniques. The resulting kinematically deforming skinned mesh (KDSM) was shown to be beneficial for both hair animation/simulation Lee et al. (2018) and water simulation Lee et al. (2019). Here, we only utilize the most basic version of the KDSM, assigning skinning weights to its vertices so that it deforms with the underlying joints similar to a skinned body surface (alternatively, one could train a neural network to learn more complex KDSM deformations). This allows us to make a very straightforward and fair comparison between learning offsets from a skinned body surface and learning offsets from a skinned parameterization of three-dimensional space. Our experiments showed an overall reduction in error of approximately

50% (see Table 2 and Figure 8) as well as the removal of visual/geometric artifacts (see e.g. Figure 9) that can be directly linked to the usage of the body surface mesh, and thus we advocate the KDSM for further study. The neural network we trained for a particular body can also be used to infer cloth with unique wrinkle patterns on different body shapes and T-shirt sizes without retraining (see supplemental material). In order to further illustrate the efficacy of our approach, we show that the KDSM is amenable to being used with recently proposed works on texture sliding for better three-dimensional reconstruction Wu et al. (2020b) as well as in conjunction with networks that use a postprocess for better physical accuracy in the $L^\infty$ norm Geng et al. (2020) (see Figure 10).

In summary, our specific contributions are: 1) a novel three-dimensional parameterization for virtual cloth adapted from the KDSM, 2) an extension (enabling plastic deformation) of the KDSM to accurately model cloth deformation, and 3) a learning framework to efficiently infer such deformations from body pose. The mean error of the cloth predicted in Jin et al. (2020) is five standard deviations higher than the mean error of our results.

## 2 RELATED WORK

**Cloth:** Data-driven cloth prediction using deep learning has shown significant promise in recent years. To generate clothing on the human body, a common approach is to reconstruct the cloth and body jointly Alldieck et al. (2018a;b); Xu et al. (2018); Alldieck et al. (2019a;b); Habermann et al. (2019); Natsume et al. (2019); Saito et al. (2019); Yu et al. (2019); Bhatnagar et al. (2019); Onizuka et al. (2020); Saito et al. (2020). In such cases, human body models such as SCAPE Anguelov et al. (2005) and SMPL Loper et al. (2015) can be used to reduce the dimensionality of the output space. To predict cloth shape, a number of works have proposed learning offsets from the body surface Guan et al. (2012); Neophytou & Hilton (2014); Pons-Moll et al. (2017); Lahner et al. (2018); Yang et al. (2018); Gundogdu et al. (2019); Santesteban et al. (2019); Patel et al. (2020); Jin et al. (2020) such that body skinning can be leveraged. There are a variety of skinning techniques used in animation; the most popular approach is linear blend skinning (LBS) Magnenat-Thalmann et al. (1988); Lander (1998). Though LBS is efficient and computationally inexpensive, it suffers from well-known artifacts addressed in Kavan & Žára (2005); Kavan et al. (2007); Jacobson & Sorkine (2011); Le & Hodgins (2016). Since regularization often leads to overly smooth cloth predictions, additional wrinkles/folds can be added to initial network inference results Popa et al. (2009); Mirza & Osindero (2014); Robertini et al. (2014); Lahner et al. (2018); Wu et al. (2020b); Patel et al. (2020). Most recently, Patel et al. (2020) parameterized cloth as a submesh of the SMPL body mesh and decomposed cloth deformation into low-frequency and high-frequency components. However, this parameterization limits cloth to be bound by the topology of SMPL, and the high-frequency folds/wrinkles added by the network are not constrained to match those in the ground truth data. In contrast, our method allows one to predict cloth deformation independent of a predefined PCA basis, and using Geng et al. (2020) ensures that folds/wrinkles are physically consistent.

**3D Parameterization:** Parameterizing the air surrounding deformable objects is a way of treating collisions during physical simulation Sifakis et al. (2008); Müller et al. (2015); Wu & Yuksel (2016). For hair simulation in particular, previous works have parameterized the volume enclosing the head or body using tetrahedral meshes Lee et al. (2018; 2019) or lattices Volino & Magnenat-Thalmann (2004; 2006). These volumes are animated such that the embedded hairs follow the body as it deforms enabling efficient hair animation, simulation, and collisions. Interestingly, deforming a low-dimensional reference map that parameterizes high-frequency details has been explored in computational physics as well, particularly for fluid simulation, see e.g. Bellotti & Theillard (2019).

## 3 SKINNING A 3D PARAMETERIZATION

We generate a KDSM using red/green tetrahedralization Molino et al. (2003); Teran et al. (2005a) to parameterize a three-dimensional volume surrounding the body. Starting with the body in the T-pose, we surround it with an enlarged bounding box containing a three-dimensional Cartesian grid. As is typical for collision bodies in computer graphics Bridson et al. (2003), we generate a level set representation separating the inside of the body from the outside (see e.g. Osher & Fedkiw (2002)). See Figure 1a. Next, a thickened level set is computed by subtracting a constant value from the current level set values (Figure 1b). Then, we use red/green tetrahedralization as outlined in Molino

et al. (2003); Teran et al. (2005a) to generate a suitable tetrahedral mesh (Figure 1c). Optionally, this mesh could be compressed to the level set boundary using either physics or optimization, but we forego this step because the outer boundary is merely where our parameterization ends and does not represent an actual surface as in Molino et al. (2003); Teran et al. (2005a).

Skinning weights are assigned to the KDSM using linear blend skinning (LBS) Magnenat-Thalmann et al. (1988); Lander (1998), just as one would skin a co-dimension one body surface parameterization. In order to skin the KDSM so that it follows the body as it moves, each vertex $v_k$ is assigned a nonzero weight $w_{kj}$ for each joint $j$ it is associated with. Then, given a pose $\theta$ with joint transformations $T_j(\theta)$, the world space position of each vertex is given by $v_k(\theta) = \sum_j w_{kj} T_j(\theta) v_k^j$ where $v_k^j$ is the untransformed location of vertex $v_k$ in the local reference space of joint $j$. See Figure 1d. Importantly, it can be quite difficult to significantly deform tetrahedral meshes without having some tetrahedra invert Irving et al. (2004); Teran et al. (2005b); thus, we address inversion and robustness issues/details in Section 5.

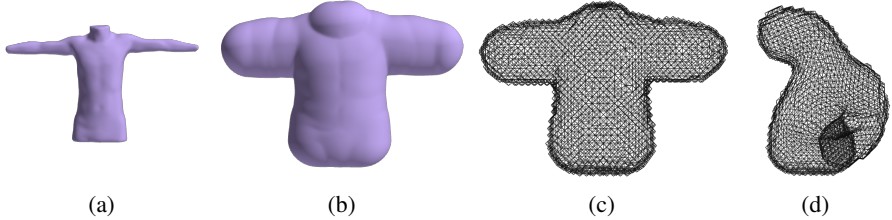

|       (a)       |       (b)       |       (c)       |       (d)       |

Figure 1: We build a tetrahedral mesh surrounding the body to parameterize the enclosed three-dimensional space. First, a level set representation of the body (a) is generated and subsequently thickened (b) to contain the clothing worn on the body. Then, we use red/green tetrahedralization Molino et al. (2003); Teran et al. (2005a) to create a tetrahedral mesh (c) from the thickened level set. This tetrahedral mesh is skinned to follow the body as it moves (d). Note that the tetrahedral mesh surrounds the whole upper body to demonstrate that this parameterization can also be used for long-sleeve shirts.

## 4 EMBEDDING CLOTH IN THE KDSM

In continuum mechanics, deformation is defined as a mapping from a material space to the world space, and one typically decomposes this mapping into purely rigid components and geometric strain measures, see e.g. Bonet & Wood (1997). Similar in spirit, we envision the T-pose KDSM as the material space and the skinned KDSM as being defined by a deformation mapping to world space for each pose $\theta$. As such, we denote the position of each cloth vertex in the material space (i.e. T-pose, see Figure 2a) as $u_i^{m_o}$. We embed each cloth vertex $u_i^{m_o}$ into the tetrahedron that contains it via barycentric weights $\lambda_{ik}^{m_o}$, which are only nonzero for the parent tetrahedron's vertices. Then, given a pose $\theta$, a cloth vertex's world space location is defined as $u_i(\theta) = \sum_k \lambda_{ik}^{m_o} v_k(\theta)$ so that it is constrained to follow the KDSM deformation, assuming linearity in each tetrahedron (see Figure 2b). Technically, this is an indirect skinning of the cloth with its skinning weights computed as a linear combination of the skinning weights of its parent tetrahedron's vertices, and leads to the obvious errors one would expect (see e.g. Figure 3, second row).

The KDSM approximates a deformation mapping for the region surrounding the body. This approximation could be improved via physical simulation (see e.g. Lee et al. (2018; 2019)), which is computationally expensive but could be made more efficient using a neural network. However, the tetrahedral mesh is only well suited to capture deformations of a volumetric three-dimensional space and as such struggles to capture deformations intrinsic to codimension one surfaces/shells including the bending, wrinkling, and folding important for cloth. Thus, we take further motivation from constitutive mechanics (see e.g. Bonet & Wood (1997)) and allow the cloth vertices to move in material space (the T-pose) akin to plastic deformation. That is, we use plastic deformation in the material space in order to recapture elastic deformations (e.g. bending) lost/recovered when embedding cloth into a tetrahedral mesh. These elastic deformations are encoded as a pose-dependent plastic displacement for each cloth vertex, i.e. $d_i(\theta)$; then, the pose-dependent, plastically deformed material space position of each cloth vertex is given by $u_i^m(\theta) = u_i^{m_o} + d_i(\theta)$.

Given a pose $\theta$, $u_i^m(\theta)$ will not necessarily have the same parent tetrahedron or barycentric weights as $u_i^{m_o}$; thus, a new embedding is computed for $u_i^m(\theta)$ obtaining new barycentric weights $\lambda_{ik}^m(\theta)$. Using this new embedding, the position of the cloth vertex in pose $\theta$ will be $u_i(\theta) = \sum_k \lambda_{ik}^m(\theta) v_k(\theta)$. Ideally, if the $d_i(\theta)$ are computed correctly, $u_i(\theta)$ will agree with the ground truth location of cloth vertex $i$ in pose $\theta$. The second row of Figure 4 shows cloth in the material space T-pose plastically deformed such that its skinned location in pose $\theta$ (Figure 4, first row) well matches the ground truth shown in the first row of Figure 3. Learning $d_i(\theta)$ for each vertex can be accomplished in exactly the same

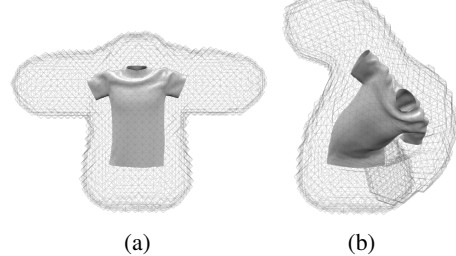

Figure 2: One can embed the cloth into the T-pose KDSM (a) and fix this embedding as the KDSM deforms (b). However, this results in undesired artifacts in the cloth (see e.g. Figure 3, second row).

fashion as learning displacements from the skinned body surface mesh, and thus we use the same approach as proposed in Jin et al. (2020). Afterwards, an inferred $d_i(\theta)$ is used to compute $u_i^m(\theta)$ followed by $\lambda_{ik}^m(\theta)$, and finally $u_i(\theta)$. Addressing efficiency, note that only the vertices of the parent tetrahedra of $u^m(\theta)$ need to be skinned, not the entire tetrahedral mesh.

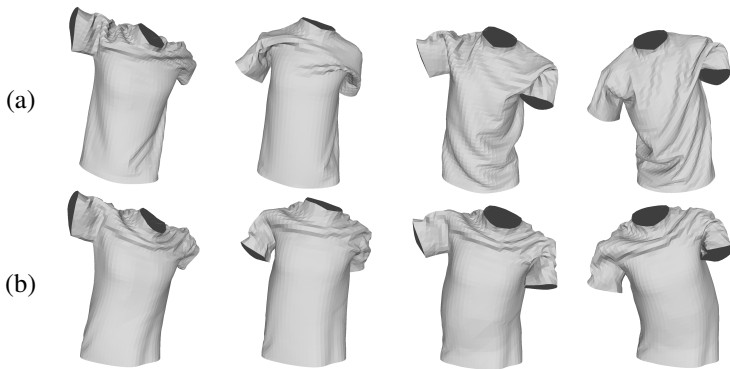

Figure 3: (a) The ground truth cloth and (b) skinning the cloth using a fixed tetrahedral embedding. Note how poorly this naive embedding of the cloth into the KDSM matches the ground truth (especially as compared to a more sophisticated embedding using our plastic deformation as shown in Figure 4).

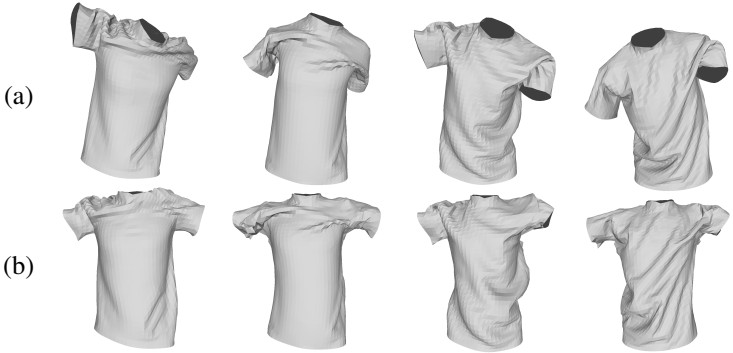

Figure 4: (a) The hybrid cloth embedding method (see Section 5) produces cloth $u(\theta)$ that closely matches the ground truth shown in the first row of Figure 3. (b) This is accomplished, for each pose, by plastically deforming the cloth in material space (the T-pose) before embedding it to follow the deformation of the KDSM.

In order to compute each training example $(\theta, d(\theta))$, we examine the ground truth cloth in pose $\theta$, i.e. $u^{GT}(\theta)$. For each cloth vertex $u_i^{GT}(\theta)$, we find the deformed tetrahedron it is located in and

compute barycentric weights $\lambda_{ik}^{GT}(\theta)$ resulting in $u_i^{GT}(\theta) = \sum_k \lambda_{ik}^{GT}(\theta)v_k(\theta)$. Then, that vertex's material space (T-pose) location is given by $u_i^m(\theta) = \sum_k \lambda_{ik}^{GT}(\theta)v_k^m$ where $v_k^m$ are the material space (T-pose) positions of the tetrahedral mesh (which are the same for all poses, and thus not a function of $\theta$). Finally, we define $d_i(\theta) = u_i^m(\theta) - u_i^{m_o}$.

## 5 INVERSION AND ROBUSTNESS

Unfortunately, the deformed KDSM will generally contain both inverted and overlapping tetrahedra, both of which can cause a ground truth cloth vertex $u_i^{GT}(\theta)$ to be contained in more than one deformed tetrahedron, leading to multiple candidates for $u_i^m(\theta)$ and $d_i(\theta)$. Although physical simulation can be used to reverse some of these inverted elements Irving et al. (2004); Teran et al. (2005b) as was done in Lee et al. (2018; 2019), it is typically not feasible to remove all inverted tetrahedra. Additionally, overlapping tetrahedra occur quite frequently between the arm and the torso, especially because the KDSM needs to be thick enough to ensure that it contains the cloth as it deforms.

Before resolving which parent tetrahedron each vertex with multiple potential parents should be embedded into, we first robustly assemble a list of all such candidate parent tetrahedra as follows. Given a deformed tetrahedral mesh $v(\theta)$ in pose $\theta$, we create a bounding box hierarchy acceleration structure Hahn (1988); Webb & Gigante (1992); Barequet et al. (1996); Gottschalk et al. (1996); Lin & Gottschalk (1998) for the tetrahedral mesh built from a slightly thickened bounding box around each tetrahedron. Then given a ground truth cloth vertex, $u_i^{GT}(\theta)$, we robustly find all tetrahedra containing (or almost containing) it using a minimum barycentric weight of $-\epsilon$ with $\epsilon > 0$. We prune this list to remove tetrahedra that may be subject to numerical precision errors that could cause a vertex to erroneously be identified as inside multiple or no tetrahedra. This is done by first sorting the tetrahedra on the list based on their largest minimum barycentric weight, i.e. preferring tetrahedra the vertex is deeper inside. Starting with the first tetrahedron on the sorted list, we identify the face across from the vertex with the smallest barycentric weight and prune all of that face's vertex neighbors (and thus face/edge neighbors too) from the remainder of the list. Then, the next (non-deleted) tetrahedron on the list is considered, and the process is repeated, etc.

**Method 1:** Any of the parent tetrahedra that remain on the list may be chosen to obtain training examples with zero error as compared to the ground truth, although different choices lead to higher/lower variance in $d(\theta)$ and thus higher/lower demands on the neural network. To establish a baseline, we first take the naive approach of randomly choosing $u_i^m(\theta)$ when multiple candidates exist. This can lead to high variance in $d(\theta)$ and subsequent ringing artifacts during inference. See Figure 5.

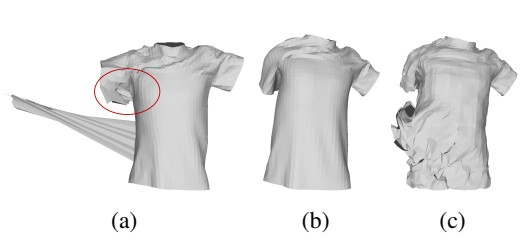

(a)    (b)    (c)

Figure 5: (a) shows a training example where overlapping tetrahedra led to cloth torso vertices being embedded into arm tetrahedra, resulting in high variance in $d(\theta)$. Although there are various ad hoc approaches for remedying this situation, it is difficult to devise a robust strategy in complex regions such as the armpit. (b) shows that the ground truth $u^{GT}(\theta)$ is still correctly recovered in spite of this high variance in $u^m(\theta)$ and $d(\theta)$; however, (c) shows that this high variance leads to spurious ringing oscillations during subsequent inference.

**Method 2:** Aiming for lower variance in the training data, we leverage the method of Jin et al. (2020) where UV texture space and normal direction offsets from the skinned body surface are calculated for each pose $\theta$ in the training examples. These same offsets can be used in any pose, since the UVN coordinate system is still defined (albeit deformed) in every pose. Thus, we utilize these UVN offsets in our material space (T-pose) in order to define $u^m(\theta)$ and subsequently $d(\theta)$. In particular, given the shrinkwrapped cloth in the T-pose, we apply UVN offsets corresponding to pose $\theta$. Although this results in lower variance than that obtained from Method 1, the resulting $d(\theta)$ do not exactly recover the ground truth cloth $u^{GT}(\theta)$. See Figure 6.

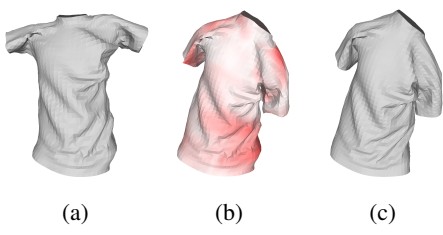

(a)      (b)      (c)

Figure 6: (a) shows the result obtained using Method 2 to compute $u_m(\theta)$ in material space (the T-pose) for a pose $\theta$. (b) shows the result obtained using this embedding to compute $u(\theta)$ as compared to the ground truth $u^{GT}(\theta)$ (c). Although the variance in $u^m(\theta)$ and $d(\theta)$ is lower than that obtained using Method 1, the training examples now contain errors (shown with a heat map) when compared to the ground truth.

**Hybrid Method:** When a vertex has only one candidate parent tetrahedron, Method 1 is used. When there is more than one candidate parent tetrahedron, we choose the parent that gives an embedding closest to the result of Method 2 (in the T-pose) as long as the disagreement is below a threshold (1 cm). As shown (for a particular training example) in Figure 7a, this can leave a number of potentially high variance vertices undefined. Aiming for smoothness, we use the Poisson morph from Cong et al. (2015) to morph from the low variance results of Method 2 to the partially-defined cloth mesh shown in Figure 7a, utilizing the already defined/valid vertices as Dirichlet boundary conditions. See Figure 7b. Although smooth, the resulting predictions may contain significant errors, and thus we only validate those that are within a threshold (1 cm) of the results of Method 2. See Figure 7c. The Poisson equation morph guarantees smoothness, while only utilizing the morphed vertices close to the results of Method 2 limits errors (as compared to the ground truth) to some degree. This process is repeated until no newly newly morphed vertices are within the threshold (1 cm). At that point, the remaining vertices are assigned their morphed values despite any errors they might contain. See Figure 7d.

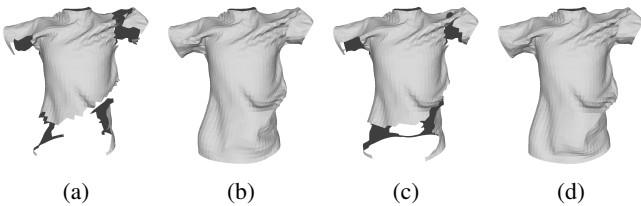

(a)      (b)      (c)      (d)

Figure 7: (a) Subset of vertices for which some choice of a parent tetrahedron using Method 1 reasonably agrees with Method 2. (b) The rest of the mesh can be filled in with the 3D morph proposed in Cong et al. (2015). (c) Subset of vertices from (b) that reasonably agree with Method 2. (d) Final result of our hybrid method (after repeated morphing).

## 6 EXPERIMENTS

**Dataset Generation:** We use the cloth dataset from Jin et al. (2020), which consists of T-shirt meshes corresponding to about 10,000 poses for a particular body Wu et al. (2020a). For each pose, the cloth was simulated on the scanned body, taking into account gravity, elastic and damping forces, and collision, contact and friction forces. We applied an 80-10-10 split to obtain training, validation, and test datasets, respectively. Table 1 compares the maximum $L^2$ and $L^\infty$ norms as compared to the ground truth for each of the three methods used to generate training examples. While Method 1 minimizes cloth vertex errors, the resulting $d(\theta)$ contains high variance. Method 2 has significant vertex errors,

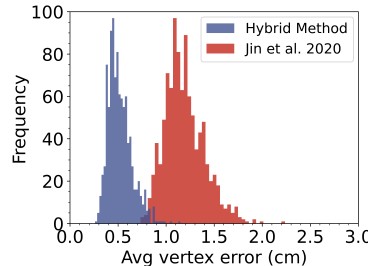

Figure 8: Histogram of average vertex errors over every example in the test dataset.

but significantly lower variance in $d(\theta)$. We leverage the advantages of both using the hybrid method.

**Network Training:** We adapt the network architecture from Jin et al. (2020) for learning the displacements $d(\theta)$, i.e. by storing the displacements $d(\theta)$ as pixel-based cloth images for the front and back sides of the T-shirt. Given joint transformation matrices of shape $1 \times 1 \times 90$ for pose $\theta$, the network applies transpose convolution, batch normalization, and ReLU activation layers. The

| Method | Max Vertex Error | Avg Vertex Error | Max $||\Delta d||$ | Avg $||\Delta d||$ |
|---|---|---|---|---|
| Method 1 | $8.9 \times 10^{-6}$ | $9.8 \times 10^{-7}$ | 136.5 | 9.35 |
| Method 2 | 12.7 | 0.549 | 14.9 | 0.75 |
| Hybrid Method | 11.6 | 0.021 | 14.7 | 0.79 |

Table 1: Dataset generation analysis (in cm). To measure variance in $d(\theta)$, we calculate the change in $d(\theta)$ between any two vertices that share an edge in the triangle mesh, denoted by $\Delta d(\theta)$.

output of the network is $128 \times 128 \times 6$, where the first three dimensions represent the predicted displacements for the front side of the T-shirt, and the last three dimensions represent those for the back side. We train with an $L^2$ loss on the difference between the ground truth displacements $d(\theta)$ and network predictions $\hat{d}(\theta)$, using the Adam optimizer Kingma & Ba (2014) with a $10^{-3}$ learning rate in PyTorch Paszke et al. (2017).

**Network Inference:** From the network output $\hat{d}(\theta)$, we define $\hat{u}^m(\theta) = u^{m_\circ} + \hat{d}(\theta)$, which is then embedded into the material space (T-pose) tetrahedral mesh and subsequently skinned to world space to obtain the cloth mesh prediction $\hat{u}(\theta)$. Table 2 summarizes the network inference results on the test dataset (not used in training). While all three methods detailed in Section 5 outperform the method proposed in Jin et al. (2020), the hybrid method achieved the lowest average vertex error and standard deviation. Figure 8 shows histograms of the average vertex error over all examples in the test dataset for the hybrid method and Jin et al. (2020). Note that the mean error of Jin et al. (2020) is five standard deviations above the mean of the hybrid method. Table 3 shows the errors in volume enclosed by the cloth (after capping the neck/sleeves/torso).

| Network | Vertex Error |
|---|---|
| Jin et al. (2020) | $1.19 \pm 0.20$ |
| KDSM (Method 1) | $1.06 \pm 0.63$ |
| KDSM (Method 2) | $0.78 \pm 0.17$ |
| KDSM (Hybrid) | $0.52 \pm 0.12$ |

Table 2: Test dataset, average vertex errors (cm).

| Network | Volume Error |
|---|---|
| Jin et al. (2020) | $2991 \pm 715$ |
| KDSM (Hybrid) | $194 \pm 161$ |

Table 3: Test dataset, average volume errors (cm$^3$).

There are significant visual improvements as well, see e.g. Figure 9. In addition, we evaluate the hybrid method network on a motion capture sequence from cmu and compare the inferred cloth to the results in Jin et al. (2020). The hybrid method is able to achieve greater temporal consistency; see the supplemental video. To demonstrate the efficacy of our approach in conjunction with other approaches, we apply texture sliding from Wu et al. (2020b) and the physical post process from Geng et al. (2020) to the results of the hybrid method network predictions, see Figure 10.

## 7 DISCUSSION

In this paper, we presented a framework for learning cloth deformation using a volumetric parameterization of the air surrounding the body. This parameterization was implicitly defined via a tetrahedral mesh that was skinned to follow the body as it animates, i.e. KDSM. A neural network was used to predict offsets in material space (the T-pose) such that the result well matched the ground truth after skinning the KDSM. The cloth predicted using the hybrid method detailed in Section 5 exhibits half the error as compared to state-of-the-art; in fact, the mean error from Jin et al. (2020) is five standard deviations above the mean resulting from our hybrid approach. Our results demonstrate that the KDSM is a promising foundation for learning virtual cloth and potentially for hair and solid/fluid interactions as well. Moreover, the KDSM should prove useful for treating cloth collisions, multiple garments, and interactions with external physics.

The KDSM intrinsically provides a more robust parameterization of three-dimensional space, since it contains a true extra degree of freedom as compared to the degenerate co-dimension one body surface. In particular, embedding cloth into a tetrahedral mesh has stability guarantees that do not exist when computing offsets from the body surface. See Figure 11. We believe that the significant

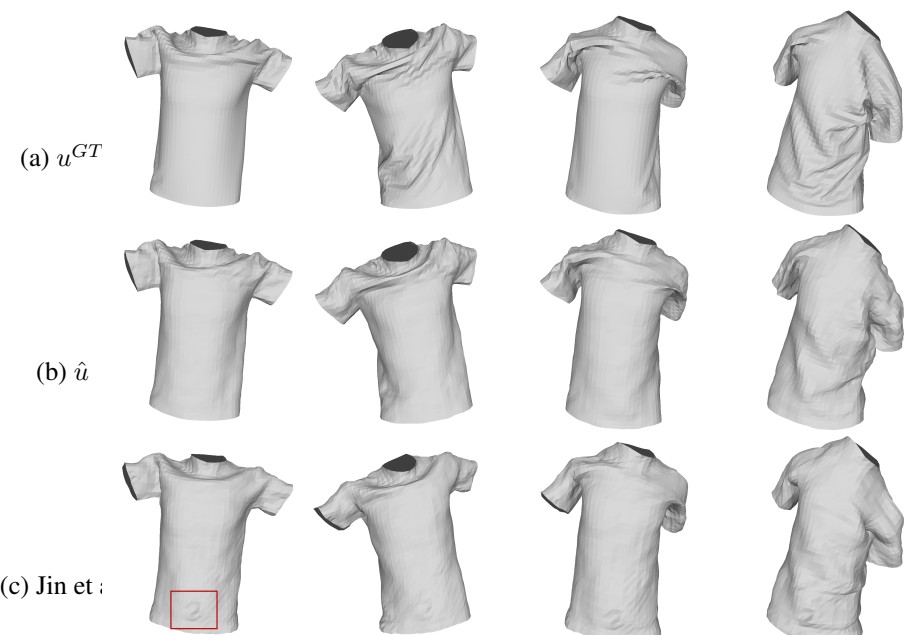

(a) $u^{GT}$

(b) $\hat{u}$

(c) Jin et al.

Figure 9: Test dataset example predictions (b) compared to the ground truth cloth in (a) and the results from Jin et al. (2020) in (c). Regularization can smooth the body surface offsets predicted using Jin et al. (2020) and as such reveals the underlying body shape, e.g. the belly button (indicated with a red square).

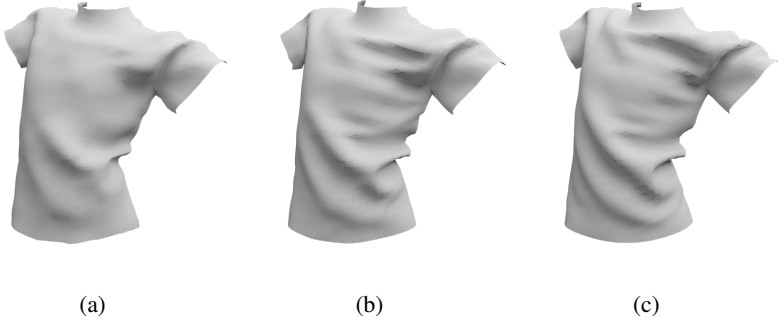

(a)                    (b)                    (c)

Figure 10: Given the hybrid method network prediction in (a), we apply texture sliding from Wu et al. (2020b) and the physics postprocess from Geng et al. (2020) as shown in (b), compared to the ground truth (c). The shown example is the same as in Figure 14 of Wu et al. (2020b).

decrease in network prediction errors is at least partially attributable to increased stability from using a volumetric parameterization.

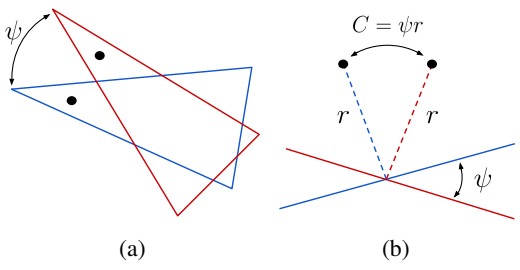

(a)                    (b)

Figure 11: (a) Embedding cloth in a tetrahedral mesh guarantees that each transformed vertex will remain inside and thus be bounded by the displacement of its parent tetrahedron. (b) However, no such bounds exist when the cloth is defined via UVN offsets from the body surface, since angle perturbations of the surface cause the cloth to move along an arclength $C = \psi r$ where even small $\psi$ can lead to large $C$ for large enough $r$.

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

## A    MODIFIED BODY SHAPES AND CLOTHING SIZES

We demonstrate that our network trained to infer cloth on a particular body, e.g. from Wu et al. (2020a), can be used for other body parameterizations and body shapes *without retraining*. Using the trained hybrid method network (see Section 6), the inferred T-shirt for a given pose is transferred to the SMPL body model Loper et al. (2015) as follows. First, we generate a skinned KDSM for the SMPL body as described in Section 3. Next, we transfer the T-pose cloth mesh to the SMPL body in the T-pose via quasistatic simulation. Then, for any skeletal pose, KDSM embedding offsets for the cloth on the SMPL body are inferred using the trained network. See Figure 13. The cloth can also be scaled to different sizes depending on user preference. See Figure 14.

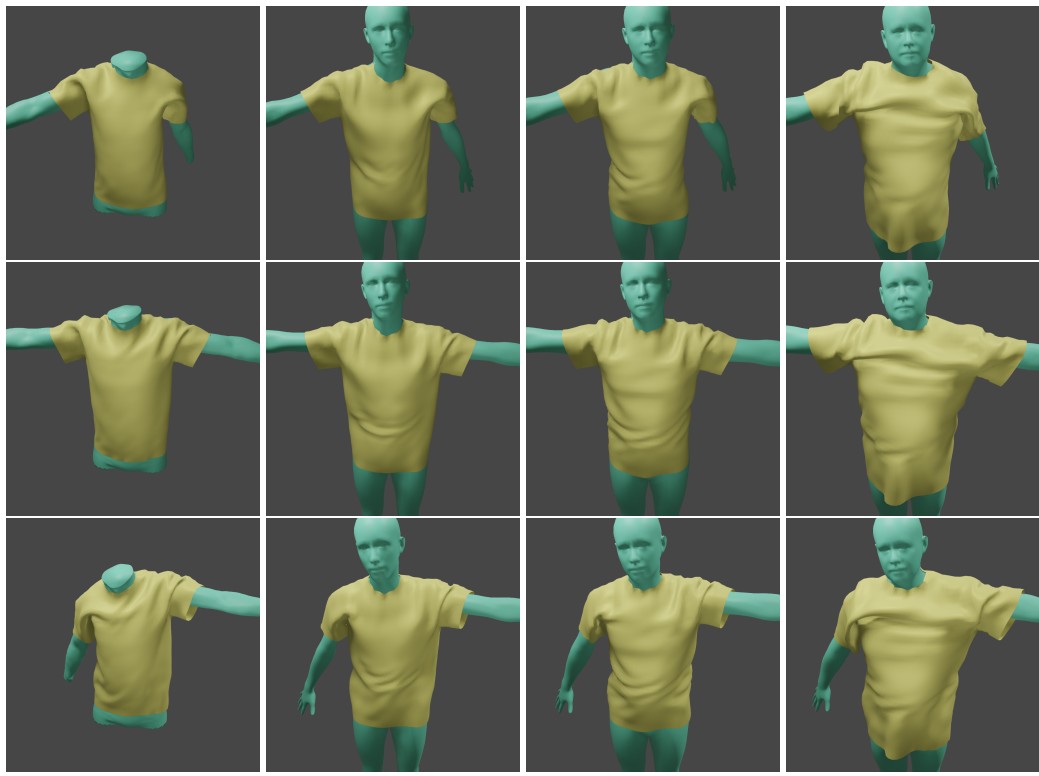

Figure 13: The network inferred cloth for the body from Wu et al. (2020a) can be transferred to the SMPL body model with any given pose and shape. Column 1 corresponds to Wu et al. (2020a), and columns 2-4 correspond to thinner, template, and thicker SMPL bodies, respectively. Note that the cloth exhibits unique wrinkling patterns depending on body shape, as expected.

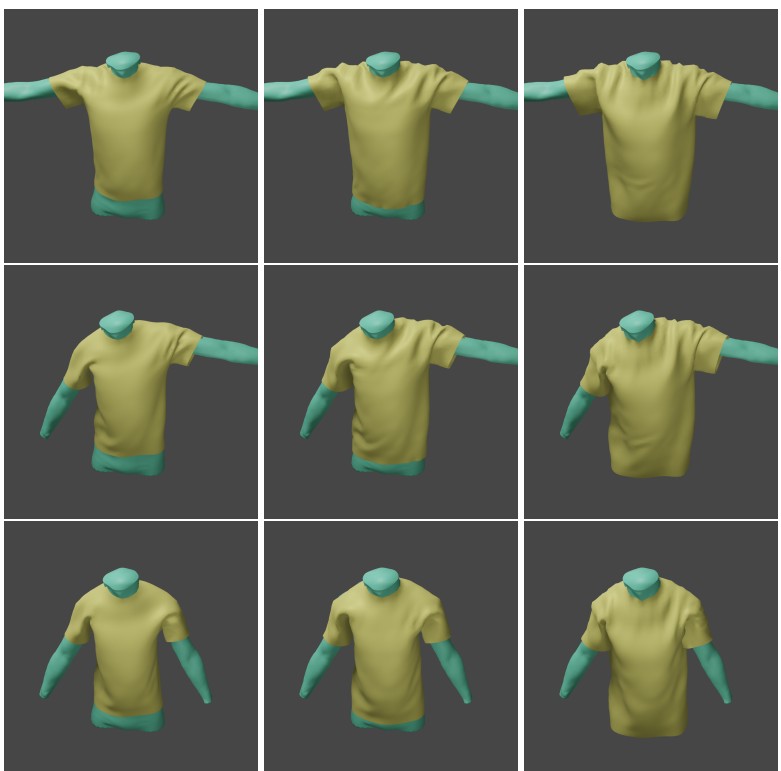

Figure 14: The network inferred cloth can be resized based on user preference without network retraining. The size of the T-shirt increases from left to right for three different poses.

