# OpenReview forum: "Skinning a Parameterization of Three-Dimensional Space for Neural Network Cloth"
_ICLR.cc/2021/Conference — Reject_

### Official Review · AnonReviewer4 · 2020-10-28
**This paper proposes to model 3D cloth by embedding it into a tetrahedral mesh that parametrizes the volumetric region around the underlying body. The idea is mainly based on KDSM[1]. Lack of experiment makes it not very convincing.**

**Rating:** 4
**Confidence:** 4

**Review:**

This paper proposes to model 3D cloth by embedding it into kinematically deforming skinned mesh (KDSM)[1], a tetrahedral mesh that parametrizes the volumetric region around the underlying body. A KDSM can be created and deformed using a variety of skinning and simulation techniques introduced in [1]. This paper extends KDSM by enabling plastic deformation in material space (T-pose), and accurately models the cloth deformation as per-vertex offsets. Inspired by [2], this paper trains a neural network to learn the per-vertex offset as a function of body pose. Once trained, the network is able to infer the 3D cloth on a particular body. Experiments show that the proposed 3D cloth parameterization method is better than the 2D UV parameterization method used in [2].

Strengths
- This paper proposes a new approach to modeling 3D cloth deformation. Experiments demonstrate that it outperforms the UV parameterization method on modeling the per-vertex offset as a function of body pose. It has the potential to be applied to other cloth-related tasks.
- This paper successfully adapts KDSM, which is originally used for hair modeling, to clothes. The inversion and robustness issues are addressed. It would be inspiring for other researchers to apply the similar idea on other types of objects.
- This paper is clearly organized and well-written. There are sufficient technical details presented in the paper.

Weaknesses
- Some existing works parameterize cloth deformation based on SMPL, e.g., SMPL+D [3], TailorNet [4], Tex2Shape [5].  This paper lacks the comparison with these closely related methods. Therefore, it is not clear whether the proposed KDSM-based cloth parameterization method is better than existing SMPL-based methods.
- There is only experimentation on synthetic data. It is not clear how it performs on real world data like BUFF [6].
- At the end of Sec. 6, the authors claim that “the hybrid method is able to achieve greater temporal consistency”. It is not clearly which component of the method enforces temporal consistency.

Suggestion
- In Sec. 5, the authors elaborate on how to robustly handle tetrahedra inversion and overlapping when generating training examples. Being depicted in natural language, the entire process is too complicated for readers to follow. For example, the sentence “We prune this list of tetrahedra, keeping only the most robust tetrahedron near each element boundary …” makes readers wonder how the robust tetrahedron and element boundary are defined. It would be better if the authors can provide a piece of pseudocode to explain the entire process in a compact and precise manner. A graphic illustration of the key operations would also be helpful for readers to understand the process.

[1] Lee, Minjae, et al. "A skinned tetrahedral mesh for hair animation and hair-water interaction." IEEE transactions on visualization and computer graphics 25.3 (2018): 1449-1459.
[2] Jin, Ning, et al. "A pixel-based framework for data-driven clothing." In Proceedings of the 19th ACM SIGGRAPH / Eurographics Symposium on Computer Animation, volume 39. Association for Computing Machinery, 2020.
[3] Bhatnagar, Bharat Lal, et al. "Multi-garment net: Learning to dress 3d people from images." Proceedings of the IEEE International Conference on Computer Vision. 2019.
[4] Patel, Chaitanya, Zhouyingcheng Liao, and Gerard Pons-Moll. "Tailornet: Predicting clothing in 3d as a function of human pose, shape and garment style." Proceedings of the IEEE/CVF Conference on Computer Vision and Pattern Recognition. 2020.
[5] Alldieck, Thiemo, et al. "Tex2shape: Detailed full human body geometry from a single image." Proceedings of the IEEE International Conference on Computer Vision. 2019.
[6] Zhang, Chao, et al. "Detailed, accurate, human shape estimation from clothed 3D scan sequences." Proceedings of the IEEE Conference on Computer Vision and Pattern Recognition. 2017.

---

> ### Author Response · Authors · 2020-11-24
> **Thank you for your feedback**
>
> We thank the reviewer for valuable feedback on our work and comments on the strengths of our paper. We provide clarifications to specific comments and concerns below.
>
> **“Some existing works parameterize cloth deformation based on SMPL, e.g., SMPL+D [3], TailorNet [4], Tex2Shape [5]. This paper lacks the comparison with these closely related methods. Therefore, it is not clear whether the proposed KDSM-based cloth parameterization method is better than existing SMPL-based methods.”**
>
> In our supplemental material (now in the appendix of the main paper file), we apply our method to the SMPL body and demonstrate that our network can be extended to SMPL and different cloth sizes without retraining. Because SMPL-based methods require that the input body is parameterized by a predefined PCA basis, whereas the body we use is not, we cannot make a fair comparison between methods such as SMPL+D, TailorNet, and Tex2Shape.
>
> **“There is only experimentation on synthetic data. It is not clear how it performs on real world data like BUFF [6].”**
>
> The dataset we used was generated by scanning a real T-shirt and body. The BUFF dataset was generated by scanning people in clothing, which does not produce separate garment and body meshes. Thus, BUFF cannot be used as ground truth data for the problem we address: inferring cloth shape from pose.**
>
> **“At the end of Sec. 6, the authors claim that “the hybrid method is able to achieve greater temporal consistency”. It is not clearly which component of the method enforces temporal consistency.”**
>
> The hybrid method robustly resolves embedding ambiguities caused by inverted and overlapping tetrahedra. This reduces variance in d(theta) and subsequently results in more accurate network predictions. Since the hybrid method inference results are closer to the ground truth, this naturally leads to greater temporal consistency.
>
> **“In Sec. 5, the authors elaborate on how to robustly handle tetrahedra inversion and overlapping when generating training examples. Being depicted in natural language, the entire process is too complicated for readers to follow. For example, the sentence “We prune this list of tetrahedra, keeping only the most robust tetrahedron near each element boundary …” makes readers wonder how the robust tetrahedron and element boundary are defined. It would be better if the authors can provide a piece of pseudocode to explain the entire process in a compact and precise manner. A graphic illustration of the key operations would also be helpful for readers to understand the process.”**
>
> Thanks for this suggestion - we have updated the sentence mentioned as follows and hope that this clears any confusion: *“We prune this list to remove tetrahedra that may be subject to numerical precision errors that could cause a vertex to erroneously be identified as inside multiple or no tetrahedra.”*

---

### Official Review · AnonReviewer3 · 2020-10-28
**Volumetric parametrization improves over surface parametrization. Insufficient novelty and experiments.**

**Rating:** 4
**Confidence:** 3

**Review:**

The authors derive a volumetric extension of the surface parameterization approach developed by Jin et.al. Towards this, they propose to use tetrahedral parameterization using well known techniques in computer graphics community. The kinematically deforming skinned mesh (KDSM) formulation for tetrahedral parameterization is borrowed from Lee at. al.

The combination of the above techniques coupled with some heuristics to increase robustness to inversion suggest improvement over Jin et. al. This is a very niche topic and I am not confident that the general audience stands to benefit from this specific formulation for clothes. The ICLR community would benefit by demonstrating the approach on other deformations of solids/liquids and validating the generality of the approach compared to other representations beyond virtual cloth. Only comparing to Jin et. al significantly limits the scope of the paper.

The computational complexity of the approach is completely ignored. As the gains over Jin et.al. seem to stem from extending the formulation to 3d domain, the compute should be compared. This becomes important for high dimensional solid/liquid simulation.

---

> ### Author Response · Authors · 2020-11-24
> **Thank you for your feedback**
>
> We thank the reviewer for valuable feedback on our work. We provide clarifications to specific comments and concerns below.
>
> **“This is a very niche topic and I am not confident that the general audience stands to benefit from this specific formulation for clothes. The ICLR community would benefit by demonstrating the approach on other deformations of solids/liquids and validating the generality of the approach compared to other representations beyond virtual cloth. Only comparing to Jin et. al significantly limits the scope of the paper. ”**
>
> Please see our submission update comment above for details on why we believe our submission is best suited for a machine learning conference. While we apply our method to the specific application of cloth inference, our contributions are relevant to the general machine learning community because we present a general machine learning paradigm of embedding high frequency detail in low frequency embedding spaces. For other solids and liquid deformations, one may use various other possible parameterizations.
>
> **“The computational complexity of the approach is completely ignored. As the gains over Jin et.al. seem to stem from extending the formulation to 3d domain, the compute should be compared. This becomes important for high dimensional solid/liquid simulation.”**
>
> The computational complexity of training our cloth KDSM network is exactly the same as in Jin et al. 2020. In both cases, the outputs of the network are front and back RGB images corresponding to vertex displacements for the front and back of the T-shirt. Our work differs in that the displacements are from the material space KDSM embedding, rather than displacements from the body surface as in Jin et al. 2020. More generally, embedding high frequency data in a low frequency embedding space does not necessarily increase dimensionality.

---

### Official Review · AnonReviewer1 · 2020-10-28
**Seems like a good paper, but more suited for some graphics conference**

**Rating:** 6
**Confidence:** 1

**Review:**

I'm not very familiar with 3d cloth animation, so may not provide an entirely adequate evaluation. However it looks to me that this paper is more suited to the graphics conference like SIGGRAPH.

The paper present a new method for cloth deformation based on tetrahedral KDSM mesh, to make the deformation more natural the neural network is used to predict the offsets  that match the ground truth deformation. Differently from Jin et al. (2020), in this paper volumetric region of air surrounding the underlying body  is used.

The are several questions about the experimental part of the paper.

1. The paper only compares on a single dataset of the Tshirts. So it is not clear how well the model will perform on other types of  clothes: regular shirts, dresses and so on.  The paper however claims that the method could be potentially to more broad categories such as hair and fluids. Is it possible to add more comparisons on the other clothes or object types?

2. How the dataset of Tshirt meshes been obtained, is it synthetically generated? If it is synthetically generated what is the benefit of using data-driven learning method? Could the method be applied on the real world scanned meshes?

Minor issues:
- Figure 1(d) looks not intuitive, does not look like (d) is modification of (c) and (d) does not look as shirt at all. It is better to use some small modification to make it more intuitive.
- Supplementary material should go along with the paper as a set of Appendixes. Only videos and source code should go as zip archive.
- Paper contain weird green artifacts in the end of first and second pages which should be removed.
- It is hard to switch attention from Figure 3 to Figure 4, these figures should be joined.

—-------—---------------

The rebuttal did not change neither my confidence, nor my impression about the paper. So I did not change my rating, however I acknowledged that other reviews may be more proficient to judge this paper properly.

---

> ### Author Response · Authors · 2020-11-24
> **Thank you for your feedback**
>
> We thank the reviewer for valuable feedback on our work. We provide clarifications to specific comments and concerns below.
>
> **“I'm not very familiar with 3d cloth animation, so may not provide an entirely adequate evaluation. However it looks to me that this paper is more suited to the graphics conference like SIGGRAPH.”**
>
> Please see our submission update comment above for details on why we believe our submission is best suited for a machine learning conference.
>
> **“1. The paper only compares on a single dataset of the Tshirts. So it is not clear how well the model will perform on other types of clothes: regular shirts, dresses and so on. The paper however claims that the method could be potentially to more broad categories such as hair and fluids. Is it possible to add more comparisons on the other clothes or object types?”**
>
> In our supplemental material, we demonstrate that our method can be applied to different body shapes and cloth sizes. The dataset available to us is specific to T-shirts, and we leave long shirts, dresses, etc. to future work. However, our work can directly be applied to these other clothing types as any item of clothing can be embedded in the KDSM.
>
> **“2. How the dataset of Tshirt meshes been obtained, is it synthetically generated? If it is synthetically generated what is the benefit of using data-driven learning method? Could the method be applied on the real world scanned meshes?”**
>
> The T-shirt meshes we used in our paper were generated by scanning a real T-shirt and simulating the cloth for each pose via physical simulation. We have updated Section 6 to add further details on the dataset we used. Our method can directly be applied to any real world scanned T-shirt mesh.
>
> **“Figure 1(d) looks not intuitive, does not look like (d) is modification of (c) and (d) does not look as shirt at all. It is better to use some small modification to make it more intuitive.”**
>
> We would like to clarify that Figure 1(d) is the result of skinning 1(c) to a new pose from the T-pose (no shirt involved).
>
> **“Supplementary material should go along with the paper as a set of Appendixes. Only videos and source code should go as zip archive.”**
>
> Thanks for the suggestion - we have moved our written supplementary material to the end of our main paper.
>
> **“Paper contain weird green artifacts in the end of first and second pages which should be removed.”**
>
> Agreed. We removed the green artifacts throughout the paper.
>
> **“It is hard to switch attention from Figure 3 to Figure 4, these figures should be joined.”**
>
> We understand that Figure 3’s caption adds a larger distance between the images in Figures 3 and 4. However, we intentionally separate Figures 3 and 4 because Figure 3 depicts using a fixed embedding in the KDSM, whereas Figure 4 depicts using the hybrid cloth embedding method.

---

> > ### Comment · AnonReviewer2 · 2020-11-24
> > **mitunderstanding of "linear blend skinning"**
> >
> > I believe we must be using a different definition of "linear blend skinning". To me, linear blend skinning is a method that takes as input: {rest positions, weights, and transformations} and outputs {deformed positions}.
> >
> > In industry the weights are painted manually by a rigging artist using software like Maya, Blender etc.
> >
> > There have been methods for automatically computing skinning weights. (for example http://skinning.org has a discussion of those).
> >
> > Is this paper assuming that hand-painted weights are given as input? Or is this method using some automatic approach to determine skinning weights?
> >
> > Hope we can settle our different definitions so I can understand this paper's method better.

---

> > > ### Author Response · Authors · 2020-11-25
> > > **Clarification on skinning weights**
> > >
> > > The inputs and outputs you stated for linear blend skinning are correct. The dataset we use includes skinning weights for the body. Those skinning weights were first generated automatically in Blender and then refined by hand. So the skinning weights are a combination of an automatic approach as well as hand refinement. Hope this clarifies any misunderstanding.

---

### Official Review · AnonReviewer2 · 2020-10-29
**Reject**

**Rating:** 3
**Confidence:** 4

**Review:**

This paper proposes a method to learn the cloth deformation of a t-shirt given
the skeletal pose of an upper body. The method skins a thick tetrahedral mesh to
the skeleton and embeds the t-shirts cloth within. At inference time a network
predicts a rest pose displacement before conducting skinning via barycentric
lookup in the tet mesh. The method is trained (as far as I can tell) on some
groundtruth cloth simulation method (this is not revealed).

I recommend rejecting this paper from ICLR 2021 on several grounds: 1) the
results are poor, 2) the description is hard to follow, 3) the methodological
choices are not well motivated, 4) the method as written is not reproducible,
and 5) the claims are too general.

1) In terms of topic and methodology this paper would be an appropriate
submission to SIGGRAPH or SCA. Callibrating for the expected result quality at
either venue, I would recommend acceptance. Since the machine learning component
of this paper is not a contribution besides being an application of
"off-the-shelf" tools, I do not see reason to lower these standards for ICLR.

2) After reading the paper, I eventually feel I understand this method.
Important details are left out effecting replicability (see below). The paper
does not clearly state what the input and output is. The paper does not describe
how groundtruth data is generated.

"This is done by first sorting the tetrahedra on the list based on their largest
minimum barycentric weight, i.e. preferring tetrahedra the vertex is deeper
inside" I don't understand this. Barycentric weights are largest when near a
vertex. Meanwhile tetrahedra can be very pancake/sliver-shaped, so that
regardless of the barycentric coordinates, a point is never deep inside. Using
barycentric coordinates to measure depth of penetration is misguided.

I didn't understand "method 2". In that method is the tet mesh entirely ignored?

Is Figure 4 showing training data or withheld testing poses?

3) The paper immediately jumps into the idea of embedding the cloth of a t-shirt
in a bulbous tetrahedral mesh around the upper body. This isn't questioned until
later when all sorts of issues appear due to overlapping and inverted elements.
There was no reason to think that skinning such a thick tet mesh was a good idea
in the first place. So the "INVERSION AND ROBUSTNESS" section is describing ad
hoc heuristics to a problem that could have been avoided by starting with a more
sound premise.

The working premise is that pose space deformations can be used for efficient
cloth simulation. This is reasonable and traces its heritage to "A powell
optimization approach for example-based skinning in a production animation
environment," which should probably be cited. From there, the choice of using a
thick tet mesh comes without solid motivation. Why not, for example, instead
learn the skinning weights and displacements directly? So, that for a point p on
the cloth the final deformation is:

∑ wi(p,θ) Ti(θ) (p + d(θ,p))

?

To generalize across body types etc., rather than the heavy handed proposed
approach of sharing this mysteriously skinned tet mesh, the learned w and d
function could be predicted based on some relative position to the rest pose and
t-shirt size etc.

4) This paper is far from replicable. How is the groundtruth data computed? Some
cloth simulation? Which method? Are collisions handled in that method?

Why does the surface boundary in Figure 1 (c) look so spiky yet the input level
set is smooth? This does not appear in the results of the red/green
tetrahedralization results. This looks more like a simply clipped regular grid
tetrahedralization.

How are the weights wkj determined? Manually? Automatically? Optimized during
training? This appears to be crucial to the method but left out.

5) Finally, this paper claims to provide a "Skinning a parameterization of
three-dimensional space for neural network cloth". Even if I accepted this paper
as successful in its results (I do not), then this paper could at best claim to
have skinned a parameterization of t-shirt deformations for upper-body motions.
The fragility of the method as discussed above makes this overclaiming
especially dubious.

---

> ### Author Response · Authors · 2020-11-24
> **Thank you for your feedback**
>
> We thank the reviewer for valuable feedback on our work. We provide clarifications to specific comments and concerns below.
>
> **“1. In terms of topic and methodology this paper would be an appropriate submission to SIGGRAPH or SCA. [...]”**
>
> Please see our submission update comment above for details on why we believe our submission is best suited for a machine learning conference.
>
> **“2. After reading the paper, I eventually feel I understand this method. Important details are left out effecting replicability (see below). The paper does not clearly state what the input and output is. The paper does not describe how groundtruth data is generated.”**
>
> Please refer to Section 6, “Network Training” where we describe the input and output of the network, as well as the respective dimensions. In particular, we state:
>
> *“Given joint transformation matrices of shape 1x1x90 for pose \theta, the network applies transpose convolution, batch normalization, and ReLU activation layers. The output of the network is 128x128x6, where the first three dimensions represent the predicted displacements for the front side of the T-shirt, and the last three dimensions represent those for the back side.”*
>
> We have updated Section 6 to add further details on how the dataset we used was generated.
>
> **“"This is done by first sorting the tetrahedra on the list based on their largest minimum barycentric weight, i.e. preferring tetrahedra the vertex is deeper inside" I don't understand this. [...]”**
>
> First, we are using the largest minimum barycentric weight rather than just barycentric weights in general. It is true that barycentric weights are largest when near a vertex, but we are looking for the tetrahedron such that the largest weight among the four vertices is the smallest (similar to concept of depth but notably different). Thus, we are not finding the tetrahedron with the largest Euclidean depth, but rather the tetrahedron that the vertex is most centered inside.
>
> **“I didn't understand "method 2". In that method is the tet mesh entirely ignored?”**
>
> No, the tet mesh is not ignored in Method 2. In Method 2, we apply UVN offsets to the cloth in the T-pose (material space), and subsequently deform the KDSM to generate the cloth in the specific pose (theta).
>
> **“Is Figure 4 showing training data or withheld testing poses?”**
>
> Figures 3-7 show training data examples.
>
> **“3. The paper immediately jumps into the idea of embedding the cloth of a t-shirt in a bulbous tetrahedral mesh around the upper body. This isn't questioned until later when all sorts of issues appear due to overlapping and inverted elements. [...]”**
>
> Inverted and overlapping tetrahedra arise due to skinning the mesh, but we handle such cases robustly via the hybrid method. We also note that addressing inversion and robustness is only done once when generating the training data - it is not a step during network inference.
>
> **“Why not, for example, instead learn the skinning weights and displacements directly? [...]”**
>
> The skinning weights for the tetrahedral mesh do not need to be learned. They are procedurally generated via linear blend skinning.
>
> **“4. This paper is far from replicable. How is the groundtruth data computed? Some cloth simulation? Which method? Are collisions handled in that method?”**
>
> Our paper is completely replicable - the ground truth data is from Jin et al. 2020, and we detail our network architecture and training/evaluation methods in Section 6. We also present both qualitative and quantitative results for all three methods discussed with regards to inversion and robustness.
>
> **“Why does the surface boundary in Figure 1 (c) look so spiky yet the input level set is smooth? [...]”**
>
> We do use a red green strategy for meshing. Because we are building a skinned volume around the body, there is no reason to compress it to conform to the exact shape of the thickened level set.
>
> **“How are the weights wkj determined? Manually? Automatically? Optimized during training? This appears to be crucial to the method but left out.”**
>
> Please see the second paragraph of Section 3 for details on how skinning weights are assigned to the KDSM. In particular, we use linear blend skinning.
>
> **“5. [...] Even if I accepted this paper as successful in its results (I do not), then this paper could at best claim to have skinned a parameterization of t-shirt deformations for upper-body motions. [...]”**
>
> The title of our paper summarizes our primary contribution in this paper. We demonstrate that our KDSM framework outperforms current state-of-the-art cloth shape prediction, e.g. predicting vertex offsets from the body surface, and provide both qualitative and quantitative examples. Furthermore, we demonstrate that our method can be directly extended to different body models/shapes and T-shirt sizes.

---

### Author Response · Authors · 2020-11-24
**Submission Update**

We thank all the reviewers for their constructive comments. In this paper, our primary contribution is a novel machine learning method whereby high frequency 3D detail is predicted by embedding data in a low frequency parameterization, e.g. a skinned tetrahedral mesh. Predicting high frequency detail is particularly challenging for the specific application of cloth inference where regularization often leads to overly smooth cloth predictions. With this challenge in mind, we demonstrate that training a neural network using our KDSM framework shows significant improvement over current state-of-the-art, e.g. predicting offsets from the body surface. We believe that the contributions of our paper are most suitable for a machine learning conference such as ICLR.

In accordance with reviewer feedback, we have made the following changes to our submission:
* We added a description of how the dataset from Jin et al. 2020 was generated (for full details, please see Section 6 in Jin et al. 2020). (R1, R2).
* We moved the supplemental material pdf to the appendix of the main paper (R1).
* We fixed the green citation artifacts at the end of the first page (R1).
* We updated the writing of Section 5 to more clearly explain how candidate parent tetrahedra are pruned from the generated list (R4).

---

### Decision · Program_Chairs · 2021-01-07
**Final Decision**

**Decision:**

Reject

**Comment:**

Three of the four reviewers recommend rejection; one additional reviewer considers the paper to be marginally above threshold for acceptance but is very uncertain and this is taken into account.  The AC is in consensus with the first three reviewers that this paper is not ready yet for publication.

There is concern from the reviewers that ICLR is not the right venue for this submission.  The author response in "Submission Update" does not clarify this concern.  Training a neural network to solve the problem does not automatically mean that ICLR or other ML conferences are necessarily the right venue.  Regardless, due to the many other raised concerns e.g. limited experimental results and comparisons as well as clarity,  the AC recommends rejection for this paper and resubmission at a more appropriate venue.